# Revisiting Entropy Rate Constancy in Text

**Vivek Verma**[*]        **Nicholas Tomlin**[*]        **Dan Klein**

Computer Science Division, University of California, Berkeley

{vivekverma, nicholas_tomlin, klein}@berkeley.edu

## Abstract

The uniform information density (UID) hypothesis states that humans tend to distribute information roughly evenly across an utterance or discourse. Early evidence in support of the UID hypothesis came from Genzel and Charniak (2002), which proposed an entropy rate constancy principle based on the probability of English text under $n$-gram language models. We re-evaluate the claims of Genzel and Charniak (2002) with neural language models, failing to find clear evidence in support of entropy rate constancy. We conduct a range of experiments across datasets, model sizes, and languages and discuss implications for the uniform information density hypothesis and linguistic theories of efficient communication more broadly.

## 1 Introduction

Linguistic functionalists have often claimed that language is *optimized for efficient communication* (e.g., Gibson et al., 2019). One common argument supporting theories of efficient communication is that humans communicate at or near *channel capacity* (Shannon, 1948); functionalists have argued that because interlocutors can only produce and comprehend a fixed amount of linguistic information per unit of time, and because speakers strategically arrange their utterances to convey as much information as possible, language should typically have uniform surprisal over time (Aylett, 1999; Bell et al., 2003; Aylett and Turk, 2004; Jaeger and Levy, 2007; Jaeger, 2010). Early evidence for this claim comes from Genzel and Charniak (2002), which used $n$-gram language models to argue that English documents exhibit *entropy rate constancy*. In this paper, we identify a limitation in Genzel and Charniak (2002)'s analysis and directly measure its hypothesis using neural language models. We then analyze results across a variety of datasets, languages, and models and discuss their implications for theories of efficient communication.

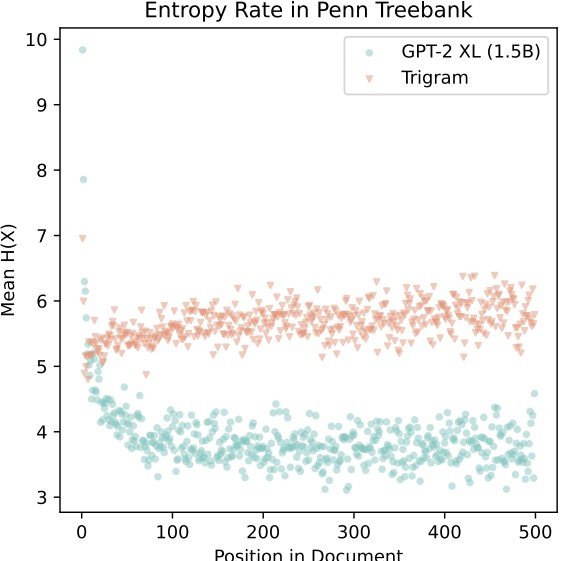

Figure 1: Entropy rate of the Penn Treebank under a smoothed trigram model and a GPT-2 XL model (Radford et al., 2019), averaged across documents per word position. Genzel and Charniak (2002) showed that entropy rate increased under $n$-gram models and predicted that it would remain constant in models which can condition on long-range context. We replicate the former result but do not find clear evidence supporting the latter.

## 2 Background

### 2.1 Efficient Communication

Claims that language is optimized for efficient communication originated with diachronic arguments about the evolution of language. Zipf (1949) observed that frequent words are usually shorter, leading to claims that word lengths are optimized for efficient communication (Piantadosi et al., 2011). Other work has argued that natural language lexicons efficiently carve up semantic space, making reference to the cross-linguistic organization of color terms (Gibson et al., 2017; Zaslavsky et al., 2018) and kinship terms (Kemp et al., 2018). Still other work has focused on syntactic efficiency, suggesting that statistical tendencies such

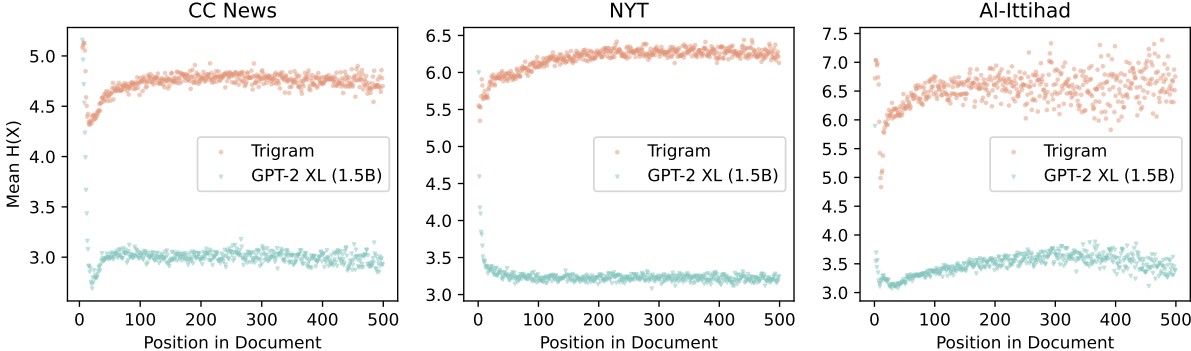

Figure 2: Entropy rate of the Common Crawl News Dataset, the NYT Annotated Corpus and the Al-Ittihad subset of the Arabic Billion Words Corpus under a smoothed trigram model and a GPT-2 model (1.5B) (Radford et al., 2019), averaged across documents at each word position. We observe a roughly increasing trend for the trigram model across all three datasets, and a variety of trends for the GPT-2 models.

as dependency-length minimization (Futrell et al., 2015), adjective ordering preferences (Hahn et al., 2018), and Greenbergian word-order correlations (Hawkins, 2009; Hahn et al., 2020) may have developed because they improve communicative efficiency. Collectively, these works demonstrate that fixed elements of linguistic structure, such as the lexicon and syntactic rules, often lead to more efficient communication than unattested alternatives.

## 2.2 Uniform Information Density

In contrast, work in psycholinguistics has highlighted the real-time decisions that speakers make in order to optimize communicative efficiency. Early work in surprisal theory (Hale, 2001) demonstrated that the contextual predictability of words determines their processing difficulty (Levy, 2008; Brouwer et al., 2010). This finding led to cognitive models of efficient communication such as the smooth signal redundancy hypothesis (Aylett and Turk, 2004) and the uniform information density hypothesis (Jaeger and Levy, 2007; Jaeger, 2010), which states that *given the choice between two otherwise identical utterances, speakers tend to choose the one with more uniform distribution of information content.* One line of evidence for the uniform information density hypothesis comes from the analysis of linguistic phenomena such as lenition (Aylett and Turk, 2004), syntactic reduction (Jaeger, 2010), and word omission (Asr and Demberg, 2015), which are more likely to appear in predictable contexts. Another line of work uses data-driven analysis of corpora to determine whether or not they exhibit properties associated with uniform information density (Genzel and Charniak, 2002,

2003; Meister et al., 2021). Crucially, research of this latter type must *operationalize* the uniform information density hypothesis in order to test its predictions; in the following section, we discuss Genzel and Charniak (2002)'s approach, which operationalized the uniform information density hypothesis at the document level.

## 2.3 Revisiting Entropy Rate Constancy

Genzel and Charniak (2002) operationalized the notion of information density by claiming that the average per-word entropy of the $n$-th sentence in English documents does not depend on $n$. In other words, they claimed that *entropy remains roughly constant over the course of documents.* Genzel and Charniak (2002) referred to this hypothesis as an entropy rate constancy principle; we use the same terminology for consistency, but we note that it differs from the standard meaning of *entropy rate* in information theory, as discussed in Section 5.

In this section, we briefly restate the argument for entropy rate constancy presented in Genzel and Charniak (2002) and refer the reader to the original paper for more details. Formally, let $X_0, \ldots, X_i$ be random variables representing words, and let $H(Y_i) = H(X_i \mid C_i, L_i)$ denote the conditional entropy of a word $X_i$ given its long-distance context $C_i = X_0, \ldots, X_{i-n}$ and a local $n$-gram context $L_i = X_{i-n+1}, \ldots, X_{i-1}$. Then by the definition of mutual information:

$$H(Y_i) = H(X_i \mid C_i, L_i) \tag{1}$$
$$= H(X_i \mid L_i) - I(X_i; C_i, L_i) \tag{2}$$

Next, assume that the entropy $H(Y_i)$ remains constant. By the above equations, mutual information

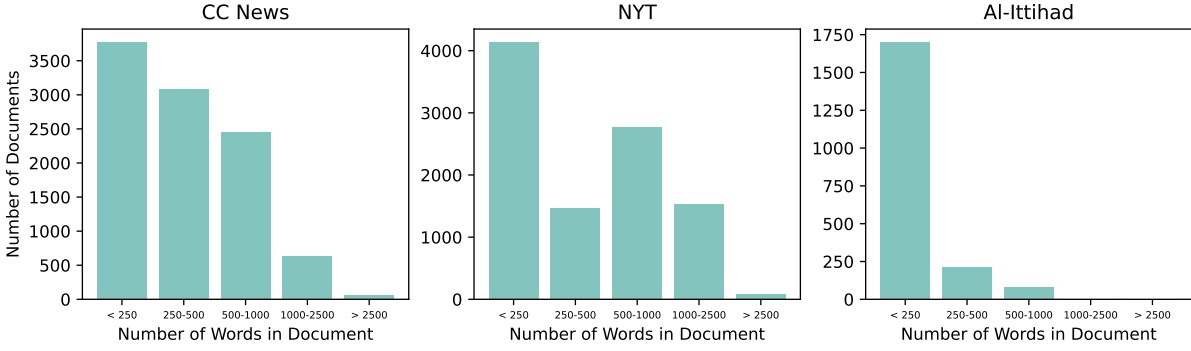

Figure 3: Distribution of document sizes for the CC News, NYT and Al-Ittihad test splits. Throughout this paper, we present word probabilities for the first 500 tokens of each article due to the lack of longer articles in each of these datasets. We provide additional results on longer documents in Appendix D.

$I(X_i; C_i, L_i)$ should increase as contexts become longer, which means that the entropy given only local contexts $H(X_i \mid L_i)$ must also increase over the course of the document.

Because Genzel and Charniak (2002) could not directly estimate $H(Y_i)$ without access to models that effectively integrate long-distance context, they instead used $n$-gram models to demonstrate that $H(X_i \mid L_i)$ increases over the course of documents. We replicate their results in Appendix 6.1 but highlight a shortcoming of their argument: non-decreasing $H(X_i \mid L_i)$ is a necessary but not sufficient condition for entropy rate constancy, and $H(Y_i)$ could increase or decrease depending on the relative value of $I(X_i; C_i, L_i)$. In other words, Genzel and Charniak (2002) confirmed a prediction of entropy rate constancy but did not provide direct evidence for the hypothesis itself. Because modern neural language models are capable of integrating long-distance contexts, we can now directly approximate $H(Y_i)$ to shine further light on these results. As shown in Section 6.2, our results do not provide clear evidence for constancy, but rather for a sharp decline at the beginnings of documents, followed by a constant or slightly declining trend.

## 3 Datasets

Genzel and Charniak (2002) ran experiments on the Penn Treebank[1] (PTB; Marcus et al., 1993), which we replicate in Section 6.1 for completeness. However, we run our primary experiments on different datasets, in order to obtain additional data with more chronological diversity, as well as non-English data. We run experiments on the

NYT Annotated Corpus[2] (Sandhaus, 2008), the Common Crawl News Dataset[3] (Hamborg et al., 2017), and the Al-Ittihad subset of the Arabic Billion Word Corpus[4] (El-Khair, 2016). We present dataset statistics in Figure 3 and describe each of these datasets, as well as our preprocessing and filtering criteria, in the following subsections.

### 3.1 The New York Times Annotated Corpus

The New York Times Annotated Corpus features over 1.8 million articles written and published by The Times from 1987 to 2007. We randomly sample 120K documents from this corpus and construct a data split consisting of 100K train articles, 10K validation articles, and 10K test articles. We condition on the title of each article when computing word probabilities and provide additional discussion of this point in Section 6.5 and Appendix A.

### 3.2 Common Crawl News

We include a subset of the Common Crawl News Dataset due to its chronological diversity. In particular, we run the majority of our experiments on GPT-2; because articles in the NYT Annotated Corpus were published between 1987 and 2007, they may appear in GPT-2's training data. To address this concern, we filtered the Common Crawl News Corpus to only include articles which were written after GPT-2 was trained. In total, there are 270996 news articles written after 2018, of which we randomly sample 100K training documents, 10K validation documents, and 10K test documents.

---

[1]https://catalog.ldc.upenn.edu/LDC99T42

[2]https://catalog.ldc.upenn.edu/LDC2008T19
[3]https://huggingface.co/datasets/cc_news
[4]https://arxiv.org/abs/1611.04033

### 3.3 Al-Ittihad (Arabic Billion Words)

Lastly, we leverage the Al-Ittihad subset of the Arabic Billion Words Corpus (El-Khair, 2016), as a means of comparing trends across languages. Although the corpus contains over three million articles, we employ one subset due to the differing nature of dialects, which would complicate comparisons. In total, we include 8551 training documents, 1K validation documents, and 2K test documents.

## 4 Models

In recent years, neural language models, in particular, transformer-based models (Vaswani et al., 2017) have been shown to greatly outperform $n$-gram models, due to their ability to scale and model long-distance dependencies. In this work, we compare the entropy rate of English text under the transformer-based GPT-2 model (Radford et al., 2019) to that of $n$-gram models.

### 4.1 Trigram Model

For each of the three datasets, we train a trigram model on their respective training splits. To provide a fair comparison to prior work, we aim to reproduce the model in Genzel and Charniak (2002) as closely as possible. However, because Genzel and Charniak (2002) did not provide exact details about its approach to $n$-gram modeling, we use parameters matching those described in the followup paper Genzel and Charniak (2003). In particular, we use a smoothed trigram model:

$$P(x_i \mid x_1...x_{i-1}) \approx P(x_i \mid x_{i-2}, x_{i-1}) \quad (3)$$
$$= \lambda_1 \hat{P}(x_i \mid x_{i-2}, x_{i-1}) \quad (4)$$
$$+ \lambda_2 \hat{P}(x_i \mid x_{i-1}) \quad (5)$$
$$+ (1 - \lambda_1 - \lambda_2)\hat{P}(x_i) \quad (6)$$

where $x_i$ corresponds to the $i$th word in a document, $\lambda_1 = 0.5$ and $\lambda_2 = 0.3$ are smoothing coefficients matching those in Genzel and Charniak (2003), and $\hat{P}$ is a maximum likelihood estimation via counts:

$$\hat{P}(x_i|x_1...x_{i-1}) = \frac{C(x_1...x_i)}{C(x_1...x_{i-1})} \quad (7)$$

where $C(x_i..x_j)$ is the number of times $x_i...x_j$ appears in the training data. We train trigram models at the word level on a closed vocabulary, as discussed in Appendix B. As a result, we note that exact probabilities may not be directly comparable to those computed by GPT-2 models, but the general trends between models are still comparable.

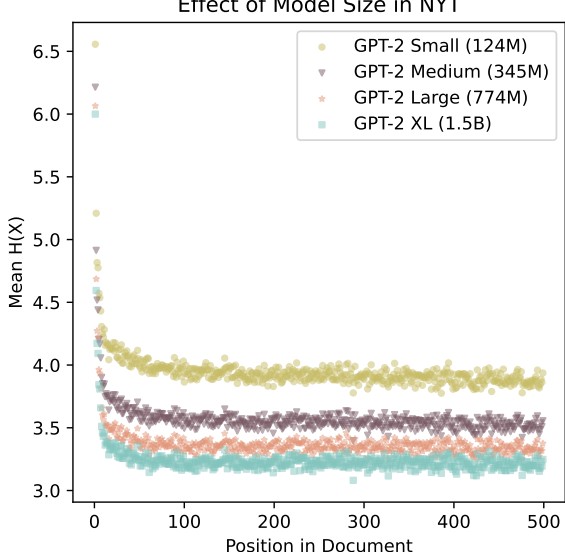

Figure 4: Entropy rate of NYT under four GPT-2 model sizes (124M, 345M, 774M, 1.5B). We note lower entropy values as model size increases but observe a consistent decline in surprisal values across all model sizes.

### 4.2 Large Language Models

In addition to training $n$-gram models, we also fine-tune GPT-2 on both the NYT and CC News datasets, with one epoch on the train split and a batch size of 8 1024-token length inputs. For fine-tuning on the Arabic Billion Words Corpus, we employ AraGPT-2 Mega (1.5B) (Antoun et al., 2021). We report results across all model sizes (124M, 345M, 774M, 1.5B), both with and without fine-tuning, in Section 6. We also report fine-tuning and inference times in Table 1 in the Appendix.

## 5 Experiments

For each dataset and model, we compute the per-token probability of each document in the dataset:

$$P_\theta(x_i \mid x_1, \ldots, x_{i-1}) \quad (8)$$

where $\theta$ denotes model parameters. We compute token probabilities using the maximum context length available to each model. Because our trigram models are trained on words and the neural models are trained on subwords, we sum over the log probabilities of subword tokens to obtain word probabilities from neural models (Mielke, 2019):

$$\log P_\theta(w_k) = \sum_{i=\text{start}(k)}^{\text{end}(k)} \log P_\theta(x_i) \quad (9)$$

for a word $w_k$ consisting of subword tokens $x_{\text{start}(k)}, \ldots, x_{\text{end}(k)}$. In contrast to Genzel and

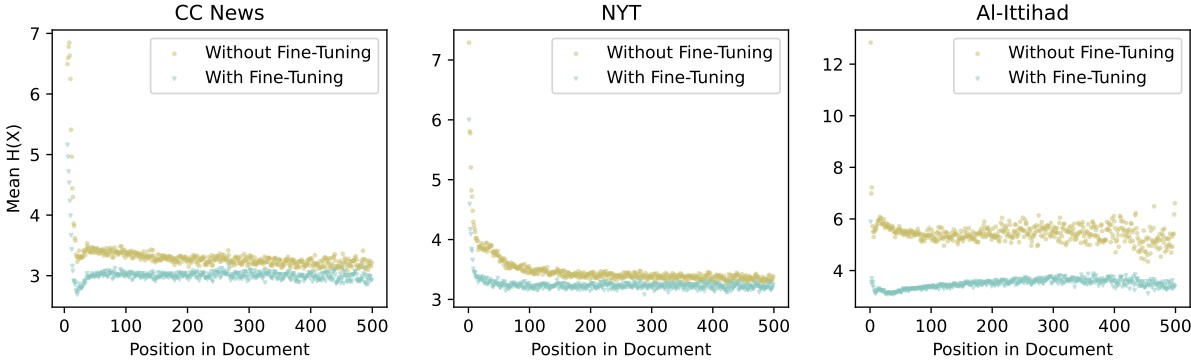

Figure 5: Entropy rate of CC News, NYT and Al-Ittihad on a fine-tuned GPT-2 XL (1.5B) compared to a non-fine-tuned model. We note that entropy values sharply decline and have lower values on the fine-tuned models, most likely due to domain adaptation. The difference between the two models is largest at the beginnings of documents.

Charniak (2002), we present results at the word-level rather than at the sentence-level to provide additional granularity and insight into trends at the beginnings of documents; additionally, this approach avoids confounding effects of sentence length which were noted in previous work (Keller, 2004; Xu and Reitter, 2018).

We then sum over each position in the dataset to compute the average per-word probability across documents in the dataset, at each word position $i$:

$$f(i) = \frac{1}{|W|} \cdot \sum_{w \in W} \log P_\theta(w_i) \qquad (10)$$

where $w$ denotes an article in a corpus $W$. Following Genzel and Charniak (2002), we refer to the slope or trend of $f(x)$ as the *entropy rate*.[5]

In this paper, we focus on a qualitative analysis of entropy rate. We avoid quantitative measures like correlation coefficients, as used in Giulianelli and Fernández (2021), because they are strongly dependent on the lengths of sampled documents. In particular: for sufficiently long documents, entropy rate must either increase or approach a constant value, because word probabilities cannot be below zero and must level off asymptotically. Meanwhile, for very short documents, we would observe

---

[5]In contrast, given a stochastic process $\{X_i\}$, Cover and Thomas (2012) defines the entropy rate $H(X)$ as the time density average entropy given by each random variable in the process, written as:

$$H(X) = \lim_{n \to \infty} \frac{1}{n} H(X_1, X_2, ..., X_n) \qquad (11)$$

While the standard definition of entropy rate refers a constant, our usage refers a more general trend over the course of documents. Further, rather than computing the limit as $n \to \infty$, we estimate the average observed word probabilities for each $i \in \{1, ..., n\}$ where $n$ is the length of the document.

a strongly negative trend, because word probability under neural models tend to decline sharply at the beginnings of documents, as discussed in the following section. We provide additional discussion of this issue, along with results from the Mann-Kendall significance test, in Appendix E.

## 6 Empirical Results

### 6.1 Replicating Genzel and Charniak (2002)

We first replicate the results of Genzel and Charniak (2002) and compare them to entropy rates achieved using GPT-2 XL (1.5B). As shown in Figure 1, entropy rates under a trigram model tend to increase, as reported in Genzel and Charniak (2002). In contrast, average word surprisals under GPT-2 XL sharply decline at the beginning of documents before leveling off. We note that these values are quite noisy, due to the test split containing only 400 documents. In the following subsections, we run similar analyses on much larger corpora.

### 6.2 Measuring Entropy Rate with GPT-2

We also replicate the results of Genzel and Charniak (2002) on significantly larger corpora, showing that trigram models exhibit increasing entropy rates on both the CC News and NYT datasets, as well as the Al-Ittihad subset of the Arabic Billion Words Corpus. We then compute entropy rate using fine-tuned GPT-2 models conditioning on the entire document history and observe various decreasing and non-monotonic trends, as shown in Figure 2. In particular, average per-word surprisal as measured by GPT-2 sharply declines at the beginning of documents in all corpora, and then either sharply rises before becoming roughly constant (CC News), asymptotically de-

clines (NYT), or slowly increases before beginning to decrease again (Al-Ittihad). This finding suggests that $I(X_i; C_i, L_i)$ and $H(X_i \mid L_i)$ do not necessarily increase at similar rates and is largely consistent with recent results about how neural language models integrate context (Khandelwal et al., 2018; O'Connor and Andreas, 2021). Most crucially, these findings do not provide clear evidence for entropy rate constancy as predicted by Genzel and Charniak (2002).

### 6.3 Effect of Model Size

We also fine-tune GPT-2 base (124M), medium (345M), and large (774M) models on the NYT dataset and observe a similar decreasing trend across all model sizes, as shown in Figure 4. As expected, across both datasets, larger models consistently exhibit lower perplexity. We predict that future large language models will continue to improve at integrating long-distance context and produce similar trends in entropy rate and provide preliminary results on GPT-3 in Appendix C.

### 6.4 Effect of Fine-tuning

Finally, we also analyze the effect of fine-tuning. We observe that fine-tuning generally results in lower surprisal values, especially at the beginning of documents, as shown in Figure 5. As a result, entropy rate tends to flatten out faster when computed with non-fine-tuned models. We hypothesize that this finding may result from *domain adaptation*: during the fine-tuning process, models may learn to attribute most of their probability mass to in-domain vocabulary and conventions. However, models without fine-tuning must determine the domain from the context alone, which may be especially difficult at the beginnings of documents.

### 6.5 Effect of Titles

In this section, we demonstrate how these results are sensitive to pre-preocessing standards. We fine-tune two GPT-2 XL (1.5B) models on CC News, one by feeding in just the document, and one with the title followed by a new-line and then the rest of the document. We compute word probabilities and only plot those corresponding to the main body of each article. Unsurprisingly, the initial word probabilities are significantly lower when conditioning on the title. However, after 100 words they are only marginally better. We note that this comparison shows that the slight increase in entropy values towards the beginning of the document seen

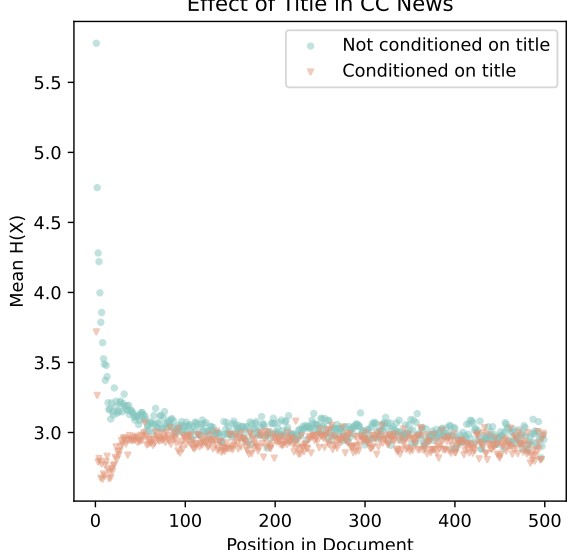

Figure 6: Entropy rate of CC News on two fine-tuned GPT-2 XL (1.5B) models, one fine-tuned with the title and one fine-tuned without. We omit entropy values for the title in this plot, but condition the model on the title.

in Figure 2 can be attributed to conditioning on the title. We hypothesize that since news titles only provide a limited amount of information, conditioning on them does not make the document significantly easier to predict. Future work might take an information-structural approach and investigate entropy rate values associated with different parts of articles, such as lede paragraphs or conclusions.

## 7 Discussion

We clarify that the findings presented in this paper do not necessarily invalidate the uniform information density hypothesis. Although entropy rate as measured by neural language models may decline over the course of documents, cognitive measures of surprisal might not decline. For example, the recently proposed lossy-context surprisal model of Futrell et al. (2020) posits that surprisal is computed with respect to an incomplete representation of context, whereas neural language models may make predictions based on lossless representations of their context windows. This perspective is also consistent with recent findings that the base GPT-2 model (124M) outperforms larger GPT-2 and GPT-3 models as a predictor of human reading time (Shain et al., 2022). In particular, these results point to a discrepancy between surprisal values under a Bayes-optimal language model and cognitively-relevant measures of surprisal. Despite still being

worse than humans at a variety of language-related tasks, we consider it likely that large language models outperform humans at the task of raw language modeling, at least as measured by perplexity. As a result, weaker language models may be better correlated with cognitive measures of surprisal.

Whether or not our results contradict the entropy rate constancy principle is a matter of interpretation. Genzel and Charniak (2002) would predict that neural language models, which are capable of integrating long-distance context, would exhibit roughly constant entropy rate over the course of documents. Under certain conditions, however, entropy rate as computed by neural language models seems to decline or even exhibit non-monotonic behavior. While this behavior is mostly isolated to the beginnings of documents, it is impossible for entropy rate to decline forever, because word probabilities cannot be less than zero. At the very least, we can conclude that our analyses do not provide clear support in favor of the entropy rate constancy principle proposed by Genzel and Charniak (2002).

## 8   Related Work

Most relevant to this work are Giulianelli et al. (2021) and Giulianelli and Fernández (2021), which explore the role of entropy rate constancy in dialogue datasets such as the Spoken British National Corpus (McEnery et al., 2017), the HCRC Map Task (Thompson et al., 1993), and PhotoBook (Haber et al., 2019). Giulianelli and Fernández (2021) follows a similar methodological procedure and computes entropy rate using fine-tuned GPT-2 models, claiming to support the entropy rate constancy hypothesis in the Penn Treebank but not in the dialogue datasets. In contrast, we focus on significantly larger news datasets, which are also more similar to the Penn Treebank data used in Genzel and Charniak (2002), and compute results across a wider range of model sizes. Using larger datasets enables additional fine-tuning and reduces variance in the results; further, our focus on word-level surprisal provides additional granularity at the beginnings of documents, where entropy rate is least constant. Finally, we note that perplexity is an extremely sensitive metric (cf. Appendix 6.5), and large variation in results may be attributable to small differences in data. In particular, we do not expect the trends we observe in news articles to always transfer to other domains, such as spoken dialogue in Giulianelli and Fernández (2021).

Recent work has also sought to connect cognitive theories of efficient communication with techniques in natural language processing; in particular, operationalizations of the uniform information density hypothesis have been connected to natural language decoding (Meister et al., 2020, 2022) and used as regularizers for language model training (Wei et al., 2021). We hope that an improved understanding of entropy rate constancy will inform such applications in the future.

## 9   Conclusion

In this work, we computed entropy rate using trigram models and GPT-2, failing to find clear evidence in support of Genzel and Charniak (2002)'s claim of entropy rate constancy in text. We provide results across various model sizes, with and without fine-tuning, and across several datasets, including the Arabic Billion Words Corpus. Our work also provides one of the only analyses of entropy rate constancy in a language besides English, although see Genzel and Charniak (2003) for results in Russian and Spanish. We encourage future work to further investigate the cross-linguistic validity of the uniform information density hypothesis.

## Limitations

One limitation of this work is that since the training data for GPT-2 was not released, it is unknown whether the contents of the NYT Annotated Corpus exist in the pre-training data. We circumvent this issue by also evaluating entropy rates on documents from the Common Crawl News dataset, filtered to those published after 2018. However, it is a possibility that time generalization may complicate the measurement of entropy rate (Lazaridou et al., 2021). Another limitation of our analysis is the sensitivity of word surprisal to small changes in text. As shown in Appendix 6.5, results can significantly change when titles are omitted from fine-tuning and inference. Handling of punctuation and other text preprocessing decisions also plays a large role in the computation of word probabilities, and consequently these decisions may affect any resultant conclusions about entropy rate constancy. Lastly, although the AraGPT-2 training data does contain the Arabic Billion Words Corpus (Antoun et al., 2021), we utlize it due to the unavailability of Arabic-based LLMs and Arabic datasets.

# Acknowledgments

Nicholas Tomlin is supported by a NSF Graduate Research Fellowship, as well as grants from the DARPA LwLL and XAI programs. We are grateful to Uriel Cohen Priva, Roma Patel, the members of the Berkeley NLP Group, as well as anonymous reviewers for feedback on earlier drafts of this paper.

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

## A  Preprocessing and Fine-tuning Details

In this section, we describe preprocessing and fine-tuning details for each of the three datasets. Fine-tuning GPT-2 across all datasets was performed with one epoch and a batch size of 8 1024-token length inputs. We outline the fine-tuning and inference times in Table 1. All experiments were run on Quadro RTX 6000 and Quadro RTX 8000 GPUs.

### A.1  NYT

We randomly sample 120K documents from the NYT Annotated Corpus and construct a training set consisting of 100K documents, a validation set consisting of 10K documents, and a test split consisting of 10K documents. We feed in each document with the title as the first line, followed by a newline (\n) token, and the body of the article afterwards. For non-finetuned runs, we replace the newline token with a colon (":").

### A.2  Common Crawl News

Similar to NYT, we randomly sample 120K documents that were written after 2018. For finetuning, we construct each document by placing the title in the first line, followed by a new line, and the rest of the document afterwards. For the non-finetuned experiments, we place the title in the first line, followed by a colon (":"), a new line, and the rest of the document after.

### A.3  Al-Ittihad

We split the Al-Ittihad subset of the Arabic Billion Words Corpus (El-Khair, 2016) into a train split, containing 8551 documents, a test split containing 2000 documents and a validation split containing 1000 documents. We then finetune AraGPT2-Mega (1.5B) (Antoun et al., 2021) by feeding in the title of each document followed by a new line, then the contents of the article after.

## B  Constructing a Closed Vocabulary

We follow additional preprocessing steps to construct a closed vocabulary for the trigram models. We first tokenize each document by splitting on whitespace and lowercasing all alphabetical characters. We then form a closed vocabulary by replacing each word which appears in the training data less than five times with the <unk> token. As a result of lowercasing and <unk>ing, we note that exact perplexity values may not be directly comparable to those computed by GPT-2 models, but

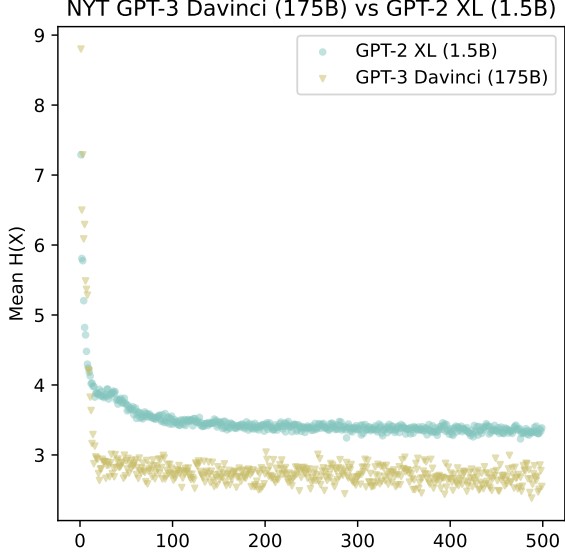

Figure 7: Entropy rate of NYT under GPT-3 (davinci) and GPT-2 XL (1.5B). Although GPT-3 perplexity values are notably lower than those of GPT-2, the general trend is similar. Neither model was fine-tuned for a direct comparison.

the general trends between models are still comparable. Indeed, we observe that the $n$-gram models are occasionally better at predicting words at the beginnings of documents, which we attribute to the frequency of rare words at the beginnings of documents which are often replaced with an <unk> token in closed-vocabulary models.

## C  Large Language Models

We primarily use GPT-2 for our experiments due to (a) its public availability, (b) the ability to fine-tune and run inference on standard hardware, and (c) the availability of comparable models for Arabic. We also provide preliminary experiments using the largest GPT-3 model (175B) but do run on all configurations due to cost considerations. We report results on 1000 documents from the NYT Annotated Corpus in Figure 7, observing a similar trend as with GPT-2 models. We use the base davinci model rather than the instruction-tuned text-davinci-003 because our work focuses on the base language modeling objective.

## D  Modeling Longer Documents

We also attempt to feed in longer documents and therefore compute entropy rates on WikiText-2 (Merity et al., 2016) using GPT-2 XL. As shown in Figure 8, these results show a non-monotonic trend

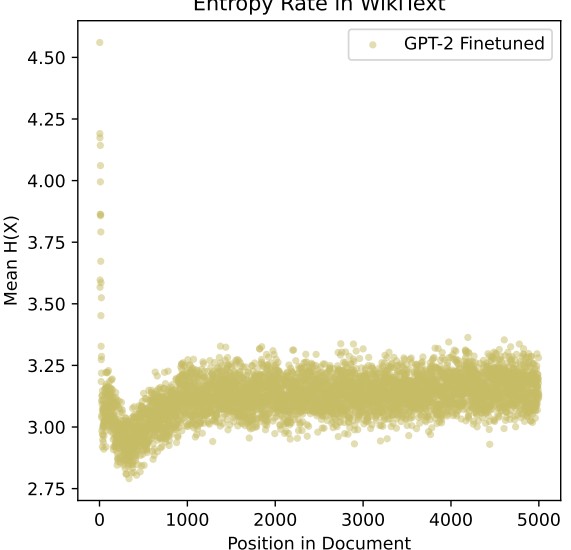

Figure 8: Entropy rate of WikiText-2 on a GPT-2 XL (1.5B), up to the first five thousand words. We run this experiment on WikiText because it has a larger number of very long documents than our primary evaluation corpora. However, we note that entropy values are not accurate due to a fixed context size and still suffer from noisiness due to a lafck of long-form documents.

even past 1000 words. We note that this result is not an accurate representation of entropy values, due to the fixed 1024 context window of GPT-2. In order to get around the fixed context window, we use a stride of 64 tokens. As a result, we expect entropy values to increase in the long run, following Genzel and Charniak (2002)'s argument for $n$-grams. We attribute the noise in these results to the lack of documents with more than 5000 words.

## E Significance Testing

We report entropy values on a per-word basis for both $n$-gram models and GPT-2. We also apply the non-parametric Mann-Kendall test (Mann, 1945; Kendall, 1948) to determine whether entropy rate is monotonically increasing or decreasing throughout the course of a document. We note that this method is not intended to compare the relative sizes of trends and that it is sensitive to hyperparameters such as the length of perplexity timeseries and choice of tokenization scheme. We omit these findings from the main body of the paper primarily due to how sensitive they are to the x-axis cutoff. We present the results and significance figures in Table 2. We further note that other methods, such as correlation coefficients or mixed effects models as used in Giulianelli and Fernández (2021) are also

highly sensitive to the length of documents, especially since entropy is least constant at the beginnings of documents. As a result of this sensitivity, we focus primarily on qualitative evaluations of the observed trends rather than on significance tests.

| Dataset | Model | Fine-tuning Time | Inference Time |
|---|---|---|---|
| CC-News | GPT-2 Small (124M) | 3 hours | 7 minutes |
| | GPT-2 Medium (345M) | 7.5 hours | 12 minutes |
| | GPT-2 Large (774M) | 15 hours | 26 minutes |
| | GPT-2 XL (1.5B) | 25 hours | 37 minutes |
| | Trigram | 7 minutes | 1 minute |
| NYT | GPT-2 Small (124M) | 3 hours | 6 minutes |
| | GPT-2 Medium (345M) | 7.5 hours | 12 minutes |
| | GPT-2 Large (774M) | 14 hours | 24 minutes |
| | GPT-2 XL (1.5B) | 24 hours | 36 minutes |
| | Trigram | 8 minutes | 1 minute |
| Al-Ittihad | AraGPT-2 Mega (1.5B) | 3 minutes | 1 minute |
| | Trigram | 1 minute | 1 minute |

Table 1: Fine-tuning and inference times across all datasets under each model size.

| Dataset | Fine-tuned | Model | Trend | p-value |
|---|---|---|---|---|
| CC News | N/A | Trigram | Increasing | $8 \cdot 10^{-8}$ |
| | Yes | GPT-2 XL (1.5B) | Decreasing | $1 \cdot 10^{-5}$ |
| | Yes | GPT-2 Large (774M) | Decreasing | 0.4 |
| | Yes | GPT-2 Medium (345M) | Decreasing | 0.01 |
| | Yes | GPT-2 Small (124M) | Decreasing | $9 \cdot 10^{-9}$ |
| | No | GPT-2 XL (1.5B) | Decreasing | $1 \cdot 10^{-17}$ |
| NYT | N/A | Trigram | Increasing | $1 \cdot 10^{-17}$ |
| | Yes | GPT-2 XL (1.5B) | Decreasing | $1 \cdot 10^{-13}$ |
| | Yes | GPT-2 Large (774M) | Decreasing | $1 \cdot 10^{-17}$ |
| | Yes | GPT-2 Medium (345M) | Decreasing | $1 \cdot 10^{-17}$ |
| | Yes | GPT-2 Small (124M) | Decreasing | $1 \cdot 10^{-17}$ |
| | No | GPT-2 XL (1.5B) | Decreasing | $1 \cdot 10^{-10}$ |
| | No | GPT-3 Davinci (175B) | Decreasing | 0.1 |
| Al-Ittihad | N/A | Trigram | Increasing | $6 \cdot 10^{-16}$ |
| | Yes | GPT-2 XL (1.5B) | Increasing | $1 \cdot 10^{-14}$ |
| | No | GPT-2 XL (1.5B) | Decreasing | $4 \cdot 10^{-7}$ |

Table 2: Results of running the Mann-Kendall test on each of the experimental conditions in this paper. In general, we observe a decreasing trend for neural models, and an increasing trend for $n$-gram models. Although this test is non-parametric, we caution that results are highly dependent on the length of the input time-series.