# OpenReview forum: "Revisiting Entropy Rate Constancy in Text"
_EMNLP/2023/Conference — EMNLP 2023 Findings_

### Official Review · Reviewer_sMtY · 2023-08-04

**Soundness:** 4

**Excitement:**

4: Strong: This paper deepens the understanding of some phenomenon or lowers the barriers to an existing research direction.

**Paper Topic And Main Contributions:**

This paper empirically evaluates the uniform information density (UID) hypothesis, originally formulated by Genzel & Charniak (2002), stating that humans tend to distribute informational content roughly in a uniform manner across an utterance. While the original claim was supported by empirical evidence from local n-gram language models, this work tests whether this pattern remains robust in light of recent neural language models (NLMs) that are able to condition on much longer contexts. In particular, authors evaluate the entropy rate for GPT-like language models over non-English and chronologically diverse datasets similar to Penn Treebank, finding evidence that goes against the original UID claim. In the discussion, authors speculate that this might be due to NLMs lossless context access compared to human memory retrieval.

**Questions For The Authors:**

A. How does your token probability computation (Eq. 8) differ from the other related papers (Giulianelli et al., 2021; Meister et al., 2021)?

**Reasons To Accept:**

* Important finding for the areas of NLG/LM evaluation, dataset/corpora analysis and specifically measuring surprisal/entropy.
* Detailed analysis featuring various data and model sizes as well as preprocessing steps, potentially sparking interesting discussions and future work in this direction.
* Very nicely contextualized within related work.
	* Similarities and differences between this and other work (mostly the papers by Giulianelli and Meister) have been thoroughly explained (§ 8, § 2).
	* To my knowledge, no relevant references are missing, which is rare in such a prolific area of work.

**Reasons To Reject:**

The analyses of the (mostly unsurprising) findings remain somewhat shallow (l. 329, l. 337, l. 357, l. 367) and there is no "deep dive" into why certain outcomes occur. I don't feel like I've learned much about how the corpora and their analyses tie into the properties of the GPT models and such takeaways are a rather crucial part of papers in this domain. This is different for e.g. Giulianelli et al. (2021) who point out the organization of context in dialogues, which can be used to "inform the development of natural language generation models." The diversity of results between corpora, as depicted by Fig. 2, might be an aspect requiring further analysis than that "these findings do not provide **clear evidence** for entropy rate constancy" (l. 321), because, from this high-level perspective, this empirical study is not very actionable. The paper often seems to cut off at points before it gets really interesting.

* One of the most surprising points to me is how entropy values do not decrease a lot after the initial dip between ~0th-50th position. This curve is even flatter for GPT-3 (Fig. 7). This kind of analysis on models with much larger parameter size (l. 773: "preliminary experiments") is very intriguing and should be fleshed out and be part of the main paper.

* The claim in l. 317-320 that the findings are consistent with how LMs integrate context needs to be more detailed. The behavior of LMs "Integrating long-range context" (l. 401) is mentioned quite often, but the paper does not really show specific results in this direction, or it should be better explained how the Figures are related to that claim.

* The interpretation of entropy values and surprisal is a bit opaque as it's currently described and depicted through Fig. 4, § 6.4, § 7.

**Reproducibility:**

4: Could mostly reproduce the results, but there may be some variation because of sample variance or minor variations in their interpretation of the protocol or method.

**Reviewer Confidence:**

3: Pretty sure, but there's a chance I missed something. Although I have a good feel for this area in general, I did not carefully check the paper's details, e.g., the math, experimental design, or novelty.

**Typos Grammar Style And Presentation Improvements:**

* The term "surprisal" is never properly introduced (l. 069, Fig. 4, l. 310).

---

> ### Author Rebuttal · Authors · 2023-08-26
>
> Thanks for your review! We’re glad you consider this an important finding and appreciated our inclusion of various datasets, model sizes, and preprocessing approaches. We’re also glad you found our presentation of related work to be thorough and accurate. In response to some specific concerns:
>
> > entropy values do not decrease a lot after the initial dip between ~0th-50th
>
> To some extent, this is inevitable (see our response to vesJ), but you are right that some models approach constancy faster than others. This is also an effect of fine-tuning (cf. Figure 5). We hypothesize that this occurs due to domain adaptation and can further elaborate on these findings in any future version of the paper.
>
> > How does your token probability computation differ from the other related papers?
>
> Our token probability computation is the same as in other work, just parametrized by different language models. The main difference is that we keep these values in terms of token probabilities rather than converting them into sentence probabilities, as in Xu & Reitter (2018) and Giulianelli and Fernandez (2021). This allows us to avoid some additional normalization based on sentence length and also provides a more fine-grained look at the beginnings of documents (256-262).
>
> > LMs integrate long range context
>
> By this, we simply mean that nearby context strongly affects word probabilities while distant context continues to weakly affect word probabilities. The cited paper by Khandelwal et al. 2018 goes into further detail on this point.
>
> As noted in our response to 8v3T, we’ll provide a clearer definition of entropy rate and surprisal in any future draft.

---

### Official Review · Reviewer_vesJ · 2023-08-06

**Soundness:** 3

**Excitement:**

4: Strong: This paper deepens the understanding of some phenomenon or lowers the barriers to an existing research direction.

**Paper Topic And Main Contributions:**

The main contribution of this paper is to investigate the uniform information density (UID) hypothesis with neural language models, and demonstrates that the original formulation of the UID hypothesis is not supported.

**Questions For The Authors:**

- Do you assume that neural language models capture not only H(X|L) but also I(X;C|L)?
- Can you employ any quantitative metrics to quantify the uniformity of information density, which can directly support the main conclusion of this paper?

**Reasons To Accept:**

- To the best of my knowledge, this is the first systematic investigation of the original formulation of the UID hypothesis across datasets (CC News, NYT, Al-Ittihad), models (Trigram, GPT-2), and languages (English, Arabic).

**Reasons To Reject:**

- As the authors themselves also point out, the main conclusion of this paper is "qualitative" or "a matter of interpretation", where no quantitative metrics demonstrate that the original formulation of the UID hypothesis is not supported. In fact, the information density looks relatively uniform to my eyes across three datasets.

**Reproducibility:**

2: Would be hard pressed to reproduce the results. The contribution depends on data that are simply not available outside the author's institution or consortium; not enough details are provided.

**Reviewer Confidence:**

4: Quite sure. I tried to check the important points carefully. It's unlikely, though conceivable, that I missed something that should affect my ratings.

**Typos Grammar Style And Presentation Improvements:**

- l.120: I(X;C,L) -> I(X;C|L)
- l.290: Figures are distributed across the paper, so that the readers should go back and forth between the pages. For example, Figure 1 on p.1 is first mentioned in l.290 on p.5.

---

> ### Author Rebuttal · Authors · 2023-08-26
>
> Thanks for your review! We’re glad that you appreciated our inclusion of several different datasets, models, and languages in this analysis.
>
> In response to your concern about the quantitative measures: unfortunately, any quantitative measure of entropy rate constancy is (necessarily) parametrized by the length of documents considered. For example: G&C ’02 chose an arbitrary cutoff at the 25-th sentence mark. Because many of the entropy rate curves in our paper are monotonically decreasing, they must eventually appear constant or nearly constant with a long enough x-axis (i.e., with long enough documents). However, it’s undeniable that there is some non-constant behavior around the first 50-100 tokens; when we say the conclusion is “a matter of interpretation,” we specifically mean that whether you consider these first 50-100 tokens important is up for debate and is ultimately a matter of how entropy rate constancy is defined. We provide a partial discussion of this issue in lines 274-285 and can elaborate further in any future version of the paper.
>
> However, note that we do actually include quantitative results from the non-parametric Mann-Kendall significance test in the appendix, which says whether a time-series is increasing, decreasing, or neither. These show an increasing trend for the trigram model and decreasing trends for almost all neural language models. However, these results come with the above disclaimer about document length and are highly dependent on our choice of x-axis cutoff (in this case, the first 500 tokens). Because of their sensitivity to document length we chose to keep them in the appendix rather than in the main body of the paper.
>
> Thanks also for your presentation suggestions. In response to your question about mutual information: we assume that LLMs directly approximate H(X | C,L) which can be used to compute the mutual information based on equation (2) in the paper.

---

### Official Review · Reviewer_8v3T · 2023-08-06

**Soundness:** 3

**Excitement:**

3: Ambivalent: It has merits (e.g., it reports state-of-the-art results, the idea is nice), but there are key weaknesses (e.g., it describes incremental work), and it can significantly benefit from another round of revision. However, I won't object to accepting it if my co-reviewers champion it.

**Paper Topic And Main Contributions:**

This paper reimagines an analysis done by Genzel and Charniak in 2002 to glean insights from statistical language models about the hypothesised constancy of information across a text. Unlike Genzel and Charniak we now have access to LLMs such as GPT which are much better predictors of language than the n-gram models of two decades ago. Taking three corpora in two languages, the authors observe the patterns in word probability across positions in a document; they do gather insights into word distribution, though they do not conclusively prove (or disprove) the entropy rate constancy hypothesis.

**Reasons To Accept:**

- It's an interesting area to study, as computational analyses of information in large corpora can inform and be informed by psycholinguistic studies.

- Since the world of language modelling has changed so much in recent years, it makes sense to revisit questions that were investigated using previous technology. Using large datasets (50 million words in each experiment so about 10x what Genzel and Charniak used) allows for more robust results.

- Studying Arabic as well as just English strengthens the generality of the insights

**Reasons To Reject:**

- The contributions are slightly unclear. The patterns found are interesting but I'm not sure what implications they have. As the authors acknowledge LLMs may be better predictors of word probability than humans, just as n-gram LMs may be worse than humans. If the psycholinguistic insights that can be achieved are limited due to this mismatch, what else can we use these results for? The paper would be stronger if the authors can help the reader see the contribution.

**Reproducibility:**

4: Could mostly reproduce the results, but there may be some variation because of sample variance or minor variations in their interpretation of the protocol or method.

**Reviewer Confidence:**

2: Willing to defend my evaluation, but it is fairly likely that I missed some details, didn't understand some central points, or can't be sure about the novelty of the work.

**Typos Grammar Style And Presentation Improvements:**

- The figures are hard to read on a non-colour printout as all the plots are similar shades of grey

- For NLP researchers without expertise in this specific area, it will be helpful to give some intuitions about how to interpret notions like entropy rate and surprisal. This seems key to following the core contribution of the paper. "High probability = low surprisal" is the way I naively think about it, but I'm still not sure how that tallies with the statements in Section 6 that "average word probabilities under GPT2-XL sharply decline at the beginning of documents before leveling off" and "average per-word surprisal as measured by GPT-2 sharply declines at the beginning of documents in all corpora", i.e. probability and surprisal decline together.

---

> ### Author Rebuttal · Authors · 2023-08-26
>
> Thanks for your review! We’re glad that you found the topic interesting and well-motivated, and that you appreciated our use of larger datasets (as well as the Arabic dataset).
>
> In response to your point about the contributions being unclear: as you say, because current LMs might be better predictors of word probability than humans, these results don’t provide *negative* evidence for UID (and in fact, it’s hard to disprove such theories). However, prior work like G&C ’02 has used LMs as *positive* evidence for UID, and we argue against this line of reasoning. This is a fairly nuanced point, but we’ll try to make it more clear in any future versions of our paper.
>
> Thanks also for the presentation tips. Our graphs are currently dual-coded (with both shape and color), but we’ll experiment with different palettes to fix the contrast issue. Your interpretation of surprisal and probability is also correct — “average word probabilities decline“ was an oversight which we will fix, and we’ll aim to provide some additional intuition about the definitions of surprisal and entropy rate in the paper.

---

### Meta-Review · Area_Chair_kc99 · 2023-09-18

**Recommendation:** 4

**Metareview:**

# Meta Review

This paper revisits the entropy rate constancy hypothesis, by Genzel and Charniak (2002), investigating it using modern language models.
They use autoregressive language models (3-grams and GPTs) to compute the average surprisal of words in a document for several document positions; they do this under a number of experimental conditions.
They then analyse whether those average surprisal values follow a constant, decreasing, or increasing trend (mostly in a qualitative manner).

The reviewers agree this is an important research question to investigate.
They also agree that this paper’s detailed analyses—including a number of models, datasets, and preprocessing steps—make a significant contribution which might spark debates and future work.
The reviewers, however, also point out that the paper lacks a deeper analysis of its results, which makes some of its contributions slightly unclear.
Overall, I agree with the reviewers that this work is interesting and that its analyses might spark future debate.
I also agree with the reviewers that a "deeper dive" into some of its results would strengthen the paper.

# Typos and Other Comments

Line 46: I believe that Zipf only observed that frequent words were shorter, and that Piantadosi et al. observed that contextually predictable sentences are shorter.

Eq 9: I think these should be log-probabilities, instead of probabilities.

Line 264: “per-word probability across”. Again, should this be log-probability instead of probability?

Eq 10: I think this is missing a $\frac{1}{|W|}$, right?

Related Work Section: I think briefly mentioning the similarities/differences between this work and Xu and Reitter (2018) here could be useful.

Line 460: “our work also provides the first analysis of entropy rate constancy in a language besides English”. See Genzel and Charniak (2003).

# References

Genzel and Charniak. 2003. Variation of Entropy and Parse Trees of Sentences as a Function of the Sentence Number.

---

### Decision · Program_Chairs · 2023-10-07

**Decision:**

Accept-Findings

**Comment:**

# Meta Review

This paper revisits the entropy rate constancy hypothesis, by Genzel and Charniak (2002), investigating it using modern language models.
They use autoregressive language models (3-grams and GPTs) to compute the average surprisal of words in a document for several document positions; they do this under a number of experimental conditions.
They then analyse whether those average surprisal values follow a constant, decreasing, or increasing trend (mostly in a qualitative manner).

The reviewers agree this is an important research question to investigate.
They also agree that this paper’s detailed analyses—including a number of models, datasets, and preprocessing steps—make a significant contribution which might spark debates and future work.
The reviewers, however, also point out that the paper lacks a deeper analysis of its results, which makes some of its contributions slightly unclear.
Overall, I agree with the reviewers that this work is interesting and that its analyses might spark future debate.
I also agree with the reviewers that a "deeper dive" into some of its results would strengthen the paper.

# Typos and Other Comments

Line 46: I believe that Zipf only observed that frequent words were shorter, and that Piantadosi et al. observed that contextually predictable sentences are shorter.

Eq 9: I think these should be log-probabilities, instead of probabilities.

Line 264: “per-word probability across”. Again, should this be log-probability instead of probability?

Eq 10: I think this is missing a $\frac{1}{|W|}$, right?

Related Work Section: I think briefly mentioning the similarities/differences between this work and Xu and Reitter (2018) here could be useful.

Line 460: “our work also provides the first analysis of entropy rate constancy in a language besides English”. See Genzel and Charniak (2003).

# References

Genzel and Charniak. 2003. Variation of Entropy and Parse Trees of Sentences as a Function of the Sentence Number.